# Political Disaffection and Digital Political Participation in Latin America: A Comparative Analysis of the Period 2008–2020

Ángel Cazorla-Martín *, Juan Montabes-Pereira and Mateo Javier Hernández-Tristán

Department of Political Science, University of Granada, 18071 Granada, Spain
* Correspondence: acazorla@ugr.es

**Abstract:** One of the issues facing the field of political behaviour analysis in recent years has been the transformation of political participation among citizens, in a context of increasing change, profoundly marked by the spread of a new digital paradigm. Network society has brought with it new forms of political participation, where different types of participatory citizenship coexist in a process of increasing interaction which, in turn, creates new morphologies, and where online and offline modes are reciprocal, generating new patterns of behaviour. Of these different types of participatory citizenship, that of the disaffected is perhaps among the most important in recent years and, in particular, since the start of the so-called "Great Recession" around 2008, and the subsequent global COVID-19 crisis. This recent context can be characterised by a significant increase in political disaffection, resulting from a loss of trust in institutions and from the constant distancing of a certain section of the citizenry from politics as a coded punishment of those governments and institutions they see as ineffective. This paper provides an analysis of citizenship types in Latin America, particularly that of the disaffected, describing their relationship to the following of political information through digital media and social networks, and identifying patterns of evolution and development in some of the trends. Results show that a clear distinction exists between the different types of citizenship and associated forms of participation, both online and offline, while also describing differences in both political perceptions and attitudes, and between areas or regions in Latin America. Likewise, important differences are found according to citizen type in relation to the following of different social networks, especially among citizens categorised as critical or disaffected.

**Keywords:** political disaffection; digital participation; social media; Latin America



## 1. Introduction

Today, the political and electoral behaviour of citizens, a subject of significant interest and ongoing study within political science, finds, in the new forms of participation enabled by the internet, and in the increase in political disaffection, two more factors to add to the existing mix of possible explanatory elements.

In relation to the former, social networks and Web 2.0 play an increasing role in citizen interconnection, social mobilisation, and political participation; they are an unstoppable force with an identity of their own which have, with their advantages and disadvantages, been instrumental in assisting and boosting democratic participation and the design and implementation of citizen initiatives [1].

In the new information and communication society, new means of communication between politicians and citizens have opened up, which are faster and more direct than more traditional means [2]. These new tools, which support citizen commitment to civic activities and political involvement [3], play an increasingly central role in the communication practices and political mobilisation of governments, political parties, and social movements [4–6].

## 2. Literature Review

Social networks and their widening use have also led to a real revolution in political communication, reconfiguring how content is produced, distributed, and consumed [7], and are leading social networks to become a key part of any political communication strategy [8]. Digital platforms offer unprecedented interactive potential in the political sphere, allowing public representatives and political parties to bypass traditional channels of communication. The mediation of journalists thus establishes bidirectional communication, which allows them to build more direct, individual relationships, to listen to people's needs, answer their questions, and encourage individual participation [9]. However, political actors tend not to exploit the full potential of social networks in terms of direct communication. In practice, social networks are primarily used as a unidirectional mode of communication [10,11]. This is the case because messages on digital platforms are often created for the purpose of being amplified by the media [12], with interactive opportunities being of secondary importance [13].

Thus, social networks and Web 2.0 not only act as communication amplifiers for political actors, but they have also led to a paradigm shift in communication. Political information is no longer the preserve of the mass media; rather, we are immersed in a new ecosystem which has fostered and driven a change of role, whereby citizens, who traditionally merely received information, have now become creators and transmitters of information as well [14]. However, it is also true that, in certain fields, very few are actually able to be heard and have influence [9].

Also among the changes brought about by social networks is an increase in the number and range of actors interacting in political communication. The advent of social networks and digital communication has seen concentrated communication, limited to that which takes place between journalists and politicians, give way to a much more open, decentralised scenario, with a greater number of actors involved [7]. So, there has been a shift from a more restricted environment to one characterised by the involvement of more actors, where much more and varied information is generated, indeed, going so far as to cause an information overload [15].

This new ecosystem does not just affect the role of citizens, but it also places social networks on an equal footing with the media as active actors in mediatisation and access to information. The mass media no longer have total control over information; rather, they now share this with social networks. In this way, users themselves can also interact directly (i) as information sources, and (ii) to influence topics of interest and the perspectives transmitted in the media. Therefore, social networks are also part of journalism in the 21st century [16].

As we have seen, while social networks have led to innovation in political information, they are not replacing journalism or the conventional media. In reality, both new communication paradigms coexist, constantly mixing and interconnecting [7,17].

Social networks have also changed the forms of communication available, which are far more numerous compared to the traditional media concentrated into just a few groups. Citizens have more options available for consuming primary information or news and, moreover, social networks are coming to dominate citizens' preferences for accessing information compared to traditional media sources (press, radio, and television), which are being increasingly relegated [7]. It is now common for a considerable section of the population to receive information only through social networks [18]. For example, in 2018, an average of 36 per cent of news consumers in 37 countries did so mainly through Facebook, with this figure rising to 48 per cent in Spain [19].

Moreover, there is also a change in citizen behaviour in relation to how they decide to access information. While traditional media requires them to exercise express will to access information, social networks offer the possibility of discovering information casually, and not always thorough an active search [7], so much so that, for almost half of citizens, the accidental discovery of news is a habitual way of getting information [19].

Political actors have long been aware that new technologies have indisputably altered the political landscape [14], especially since the internet strategy used in Barack Obama's 2008 presidential campaign in the US. For years, these technologies have played a key role, meaning that political parties also focus their strategies on these new modes of communication, using these channels of interaction with citizens to gain a wider dissemination of their messages [20].

The reason social networks have assumed a key role in political communication is based on factors such as their ability to improve knowledge of and communication with the electorate, and the fact that they allow politicians to reach the citizenry directly without input from journalists, as well as convey targeted messages to particular types of citizens [14].

In terms of which are the preeminent and most influential platforms, they are not all designed for the same purpose, nor are they aimed at influencing political participation per se. Among the most notable are Facebook and Twitter, which are the most popular for disseminating different political discourses [21]. Twitter is more appropriate for the dissemination of news and current affairs, while Facebook is aimed more at creating communities of users and, therefore, at online organisation and mobilisation. Political actors use Facebook mainly to disseminate content relating to their public engagements, campaign events and proposals, and to seek votes [11,22]. In terms of reach, the algorithms used by this platform mean that the reach of posts on pages usually used by political parties and candidates is limited to the people who already follow them [23] (Stier et al., 2018), unless they invest in advertising, which boosts impact. It is a social network that is used and valued more by local- than national-level politicians [24], since it offers the chance to interact with citizens, and allows for the mass dissemination of their message to a particular community.

Also notable is the preeminent messaging platform WhatsApp, which is aimed at the exchange of personal information, and the platforms YouTube and Instagram, which are leaders in audio-visual and photographic content, respectively, and focus more on entertainment [7]. YouTube is aimed mainly at storing and sharing videos [25], which are also played on other social networks. Of these, Instagram is the most visual network, insofar as the image takes centre stage, accompanying text is of secondary importance, and it is not possible to post links [25]. Instagram is increasingly playing a part in political communication, mainly as a way of humanising the candidate and projecting a more friendly image to the citizen [26]. The use of this platform is aimed mainly at publishing content from the private sphere [27], showing public engagements [28], or highlighting leadership qualities through activities carried out in the line of public service [8,29].

Of all the social networks, however, Twitter is without doubt the most prominent social network and a model in the field of political communication, disseminating information and fomenting debates in real-time and with global reach [30,31]. Furthermore, this platform has democratised the public conversation, since, thanks to this network, citizens can interact directly with their political representatives and vice versa, while also being able to take part directly in debates on public issues generated within the network [14]. Twitter is also the social network with the largest presence of politicians, political parties, and specialised journalists compared to other platforms, which puts it at the centre of digital political debate.

Another aspect of this network that has contributed to its position is that it allows citizens' opinions to be garnered directly and even categorised, such that journalists consider it to be a barometer of public opinion [32], particularly in relation to trending topics. Furthermore, communication on the platform can be simultaneous, differentiated, and retransmitted. In addition, communication is public and can reach all those taking part in the conversation or those who are interested in it [30]. Another characteristic is the brevity of the messages, which are limited to 280 characters[1], generating content particularly focused on micropolitics, with succinct, easily consumed ideas [33], which are particularly

important in a society where immediacy takes precedence and there is a demand for simple, succinct messages.

In terms of the popularisation of social networks and digital technologies among political parties and their regional leaders, a study by Gamir et al. [25] on the evolution of digital political communication in the Spanish general elections of 2011, 2015, and April 2019, finds a continual increase in the digital presence of the leading candidates of the main parties in the 52 constituencies: almost 8 of every 10 (78.5 per cent) actively participated through a personal website or blog, Twitter, or Facebook in the 2019 elections compared to 53.8 per cent in 2011.

Nevertheless, the use of platforms among regional political leaders has changed over time. The most notable change has been in personal blogs, which moved from being most actively used in 2011 to being barely mentioned in the 2019 elections. The study also corroborates what is widely noted in relation to Twitter: that it is the social network with the largest presence in politics. Finally, Instagram is the social network experiencing most growth, overtaking Facebook in the 2019 elections, which is itself being used less and less.

The involvement of social networks in the political behaviour of the electorate has also led to citizens participating more and being increasingly capable of backing a more socially-focused media agenda in the face of the trivialisation of information [34]. Moreover, behaviour such as consuming information and news on political or public interest topics, posting opinions, entering into public debates or offering an opinion, contacting political representatives, organisations or social movements, among others, means there is a transfer from online activity to a mobilising offline effect [35], thanks to political participation on platforms such as Facebook and Twitter [36]. This evidence is yet to be compared with the messaging platform WhatsApp [37].

With regard to the geographical context and scope of the current study, internet penetration and the use of social networks in Latin America is weak in comparison to more developed countries. Although this has increased considerably in recent years, moving from 20.9[2] (out of 100) in 2004 to 49.9 in 2018, it still lags behind other regions, such as Western Europe (with an index of 71.06), North America (80.85), Eastern Europe (52.9), and the Middle East and North Africa (55.54). Nevertheless, it is ahead of other regions, such as Africa (35.05) and the Asia-Pacific (49.16). In terms of the annual growth rate of digitalisation, Latin America saw an increase of 6.21% from 2014 to 2018, which put it behind other emerging regions, such as the Asia-Pacific (9.39%), Africa (8.27%), and Eastern Europe (6.89%). However, developed and industrialised countries saw a lower growth rate than emerging regions, with an annual growth rate of 4.28% in Western Europe, and 3.94% in North America. This is due to these regions now being in a latter phase of digital ecosystem development, with a lower capacity for growth (Development Bank of Latin America. Caracas, Venezuela. CAF 2020).

Within Latin America, levels of digitalisation vary across the region, and this variation is closely related to the economic and social development indicators, not only of each country, but also the social and economic divisions in each state. The digital gap and problems with internet access relate to variations in the telecommunications infrastructure necessary for connection, which may be limited by low population density in certain zones, by poverty, or by a lack of political will and guidelines for its development (United Nations Educational, Scientific and Cultural Organization. París, France. 2017).

### 3. Theory

Finally, the main focus of the current study is the impact of political disaffection on digital political participation and, in particular, on the following of social networks in Latin America. In this sense, beyond connectivity per se, it is important to examine the unifying points that exist between different citizen types, and the different types of participation. In this regard, there are theories that point to how this online participation transforms into offline political behaviour, influencing types of political activation more generally.

The mobilisation thesis [38] posits that the internet and social networks encourage individual political behaviour, including among citizens who have previously not participated in conventional political activity [39]. This theory also points to the reinforcement thesis, which is a more pessimistic perspective suggesting that digital political activism is predominantly used and monetised by citizens with a higher educational and economic status, since they have the digital ability and knowledge necessary to use the internet with political vision [40,41]. Finally, the "slacktivism" thesis (armchair activism) holds that using the internet and social networks for political activism does not mean this is translated directly into the mobilisation of offline political actions [42].

As already mentioned, the other major factor referred to in the shaping of citizen behaviour is political disaffection. This phenomenon is a characteristic and defining feature of the political culture of citizens of most Western societies, which not only shapes their behaviour in a given moment, but is also an enduring factor which can be transmitted from generation to generation [43], and whose intensity, although independent of context, can indeed be activated [44].

Political disaffection has influenced the behaviour of citizens in a range of contexts, particularly notable in recent years, caused by the financial and economic crisis of 2008. The impact of this was mainly felt in employment and loss of purchasing power among the population, and the associated reduction in quality of life. Yet another major effect to emerge has been the distancing between the citizenry and representative political institutions, something that has been demonstrated in Europe in the form of social protest movements, such as the 15-M in Spain (2011), the "yellow vests" movement in France (2019–2020), and Brexit in the United Kingdom (2016–2020). In the case of Latin America, we can identify the protest movements in Chile (2019–2020), the riots and protests in Ecuador (2019), and those currently taking place in Peru (2022–2023). In the same way, the COVID-19 health crisis and all its disastrous social and economic repercussions have had a significant and negative impact on citizen trust of political institutions, on how the work politicians carry out is viewed, and even on interest in politics and public service. In this way, the category of disaffected citizens is linked more to criticism and dissatisfaction with the functioning of the political system than to a structural distancing related to disinterest as such. This effect is known as "overload" [45,46] and is rooted in the comparison between citizens' personal situation (egotropic) and that of society as a whole (sociotropic), and the projection of this onto increasingly sceptical attitudes towards the system [47]. In this way, perceptions and perceived emotions have a huge influence (inputs) on citizens, and mediatisation via the mass media and social networks acts as a multiplier in the construction of collective emotions [48].

Although political disaffection has commonly been linked to attitudes such as dissatisfaction, mistrust or apathy towards politics [44], lack of interest, inefficacy, inconformity, impotence, frustration or rejection, among many others, it is true that there is an extensive academic literature in which disaffection is treated as an isolated component, with particular and different characteristics lending their own conceptual aspects [49]. Nevertheless, this does not mean that its definition and scope escape complexity, as shown by Rivera and Pereira [48] in relation to three factors: the difficulty in determining which dimensions shape it, the fact that political disaffection can be easily confused with other concepts such as disillusionment, democratic legitimacy, or electoral demobilisation, and the scant importance given by political science to the emotional sphere when explaining political disaffection. In relation to this last factor, "emotional intelligence theory" [50] stands out as a school of thought that puts emotions as the central driving force behind changes in citizens' political attitudes. In turn, these three factors bring the challenge of categorising and operationalising the concept.

Nevertheless, an initial approach to the concept of political disaffection may focus on two dimensions: structural, linked to the deactivation of mechanisms of political participation available to citizens, caused by distancing, apathy, and disinterest in public issues, and; second, the activation of purely affective components, whereby negative attitudes towards

politics are activated based on feelings such as anxiety or aversion, that are derived from an emotional construct and eventually activate this kind of disaffection [48].

More specifically, Di Palma [51] states that political disaffection involves "a subjective feeling of inefficacy, cynicism, and lack of trust in the political process, politicians and democratic institutions, but without this meaning the legitimacy of the political regime is questioned". In this way, disaffection is linked to negative attitudes in political culture that affect citizen participation in the foundations of political systems, although without questioning the suitability of the system itself [52]. This perspective brings in a dimension of evaluation of the political system and of distancing between citizens and institutions as a result of a lack of trust [53]. Other authors [54] point to a lack of interest among citizens in politics and public affairs as the cause of their inaction and lack of willingness to participate politically.

Another approach to disaffection is related to economic uncertainty, and to how citizens evaluate the economic performance of managers and its connection to the perfection of democracy, a relationship that has been widely studied in countries such as Portugal and Spain [55] and that links disaffection with the economic vote perspective [56,57].

## 4. Materials and Methods

As mentioned above, the complexity of the concept of political disaffection also relates to the problem of how to construct the dimensions needed to be able to categorise and operationalise it. A range of authors [53] opted to use lack of interest and disillusionment with politics as the dimensions that constitute and operationalise political disaffection. Others [58] also used two dimensions; in this case, interest in politics and trust in parliament. In relation to the latter, and according to the combinations in each dimension (interested/not interested in politics; trust/distrust of parliament), citizens can be classified into four categories: civies[3], deferentials, critics, and the disaffected.

The disaffected and deferentials are those citizens who show interest in politics but who, nevertheless, differ in their trust of parliament, or in the political system. While deferentials trust the system, the disaffected do not. On the other hand, critics and civies share a disinterest in politics, but differ in their position on trusting parliament, with critics showing distrust and civies trust. Lorente and Sánchez [44] simplify this categorisation by combining the categories civies and deferentials into a single category, "satisfied", in the sense that they trust institutions.

In addition, it should be noted that the relationship between political disaffection, its continuity over time or activation, and its impact on citizens' electoral behaviour [59] has not been studied in depth [44].

Magalhães [60] is one of the few authors to analyse this in greater detail, finding, in his study conducted in Portugal, that voters classified as disaffected have different levels of cognitive mobilisation and political participation, in line with Cazorla et al. [57], who show that political disaffection, alongside other factors, explained the high levels of abstention in the European elections of 2014. Furthermore, as a result of their distrust of the political system and institutions, disaffected citizens may direct their vote towards political groupings whose ideology also reflects this dissatisfaction with the system [44].

Finally, critical citizens are those whose view is based on utilitarian rationality, inasmuch as their political behaviour is mainly based on their perception of the candidates competing for election, on how they evaluate their past performance, or on how they may benefit from future promises. In other words, the critic substitutes emotions and cognitive shortcuts, such as party ideology or identification, for the evaluation and selection of those political options which best fit their interests at a given moment [61].

The current research was divided into two phases. The first comprised evolutionary-type analysis of the following of political information in Latin America via digital media and social networks, working with the data of the Latinobarometer from 2008, 2013, 2016, and 2020.

The second phase consisted of analysis of data from the latest edition of the Latinobarometer (2020), in which a typology of political citizenship has been developed, with a view to isolating disaffected citizens in Latin America. To this end, based on the prior categorisation carried out by Montero, Navarrete, and Sanz [58], we created a bidimensional variable to include the two main categories used in studies of disaffection, for which the following variables of the Latinobarometer were selected (Table 1).

**Table 1.** Citizen typology.

|  | **Not interested** | **Interested** |
| --- | :---: | :---: |
| Institutional Trust (+) | Civies | Deferentials |
| Institutional Trust (−) | Critics | Disaffected |

The outcome is a categorisation in which the four citizen types are classified: civies (disinterested and trusting), deferentials (interested and trusting), critics (disinterested and distrusting), and the disaffected (interested and distrusting), which are then—following Lorente and Sánchez [44]—reduced to three categories, whereby civies and deferentials are combined to make the category "satisfied". This classification allows us to work with a differential typology of satisfied, critical, and disaffected citizens. Finally, the last phase involved a Multiple Correspondence Analysis (MCA[4]), in which the combined relationship between citizen profiles, digital media consumption, social networks, and groups of countries in Latin America were analysed.

## 5. Results

One of the first issues of interest for analysis is how the use of the internet and social networks to follow political information in Latin America has evolved, in that it is a process that has gone hand in hand with the expansion of the digital society across most of the world. Latinobarometer data show patterns of continuity in relation to overall trends in the region, albeit with some differences. While projected data from the most recent 2020 edition in relation to this measure are fairly similar to those of other regions, such as Europe or the United States, it should be noted that it has risen much more sharply in Latin America. In 2008, a mere 11% of citizens followed political information using digital media (Figure 1), a figure which had almost quadrupled by the time of the most recent edition. Furthermore, future projections show a growing trend which will eventually find half the population actively following political information through digital media.

In terms of social network-following, there is a similar trend among most networks, especially in the case of WhatsApp, Facebook, and Instagram. Only in the case of Twitter does this general trend of explosive growth over the last 10 years show relative stagnation in terms of implementation and use. In this way, it can be seen that the main digital media used are those intended for general information-sharing and instant communication, namely WhatsApp, Facebook, and YouTube, while, as has already been seen, more specific networks, such as Instagram and Twitter, make up a second group, whose use is much more closely linked to active online political participation and information.

Similarly, these data are produced in a political and economic context that may be characterised as having high levels of political and economic scepticism, perhaps the most evident feature of the period of the so-called "Great Recession".

Analysis of data on institutional trust (derived from trust in congress or national parliament), and levels of satisfaction with democracy (derived from the democratic satisfaction scale) for all Latin American countries clearly demonstrates this context of continued erosion of political and institutional trust (Figure 2). Specifically, the proportion of citizens who trust congress or national parliament drops from 40.8% in 2008 to 26.8% in 2020, while those who trust democracy falls from an already very low 34.1%, to 20.5% over the same period. In both cases, there is a continual decrease, a trend which is incompatible with democratic stability in the region's immediate future. A similar trend can be seen in relation to how

the economic situation in Latin America has developed, especially in countries which have been particularly hard-hit by the dual effects of the economic crisis and COVID-19, and which show growing economic scepticism prevalent in the region. In this way, positive assessment of the economic situation in each country drops by 14 percentage points over the period under study, while the number of citizens assessing the economic situation as bad or very bad increases by 10 points [62].

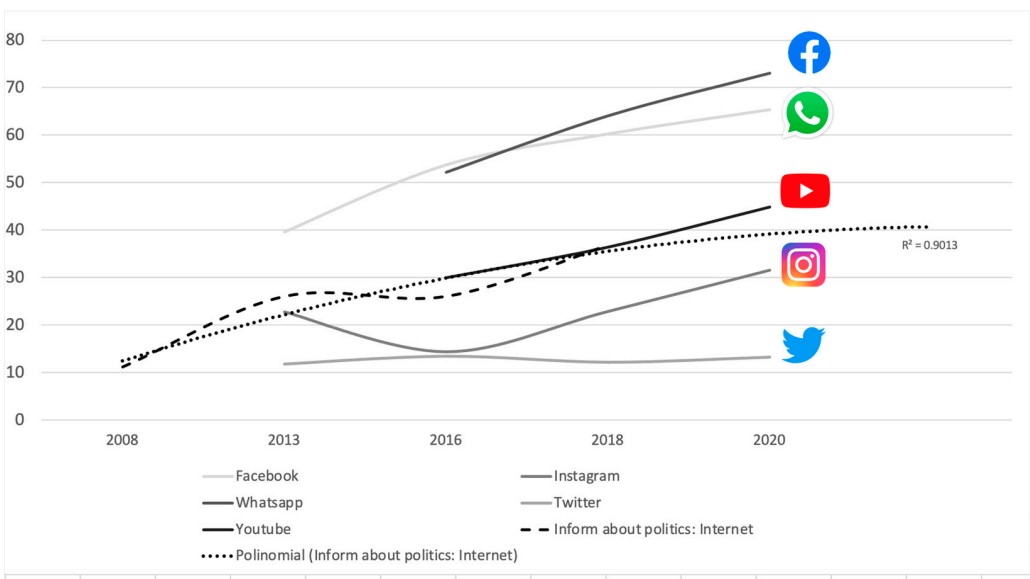

**Figure 1.** Social networks and online political information-following in Latin America. Source: Latinobarometer (2008–2020).

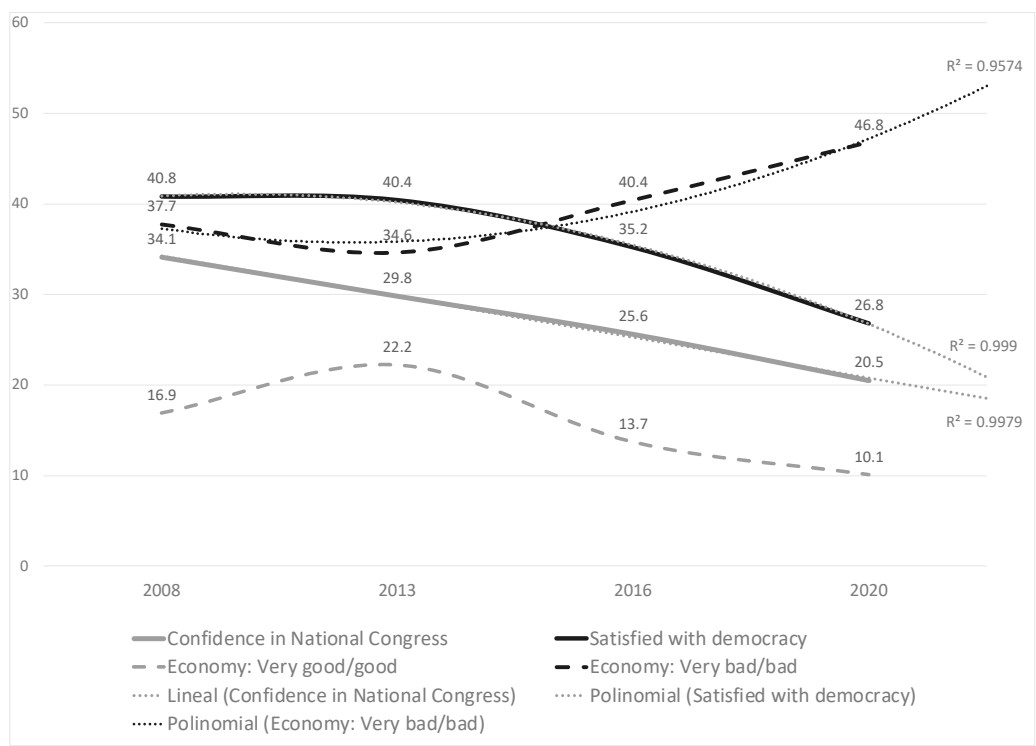

**Figure 2.** Trust in national congress and satisfaction with democracy 2008–2020. (Trend analysis using regression analysis). Source: Latinobarometer (2008–2020).

In this digital, political, and economic context, an indicator was developed related to the type of citizen. This was based on the general hypothesis that the deteriorating political and economic situation in Latin America has resulted in the upsurge of a new type of citizenship, which may be defined as critical or disaffected. This category is based on their degree of connection or lack of connection with the system, and the level of their institutional distrust. It is a form of punishment for governments and institutions seen as ineffective and incapable of providing solutions

As already outlined in the methodology section above, a variable was developed which comprised satisfied, critical, and disaffected citizen types, with results showing that the percentage of critical and disaffected citizens far exceeds that of satisfied citizens in this group of countries. It should be noted that it has not been possible to trace the evolution of this pattern, since the variable "interest in politics"—essential to be able to work with the categorisation proposed by Montero, Navarrete, and Sanz [58]—is only available for the year 2020. Specifically, 59.8% fit the profile of critical citizen, compared to 20.6% satisfied and 19.8% disaffected (Table 2). These data build on the idea developed above in relation to the weight of contextual economic, political, and social factors, and the spread of disaffected and critical types of citizenship.

**Table 2.** Citizen typology.

|  | **Satisfied** | **Critics** | **Disaffected** |
|---|---|---|---|
| Argentina | 18.8 | 50.5 | 30.7 |
| Peru | 7.5 | 62.3 | 30.2 |
| El Salvador | 12.3 | 62.7 | 25 |
| Colombia | 15.1 | 60.4 | 24.5 |
| Chile | 13.1 | 65.5 | 21.3 |
| Mexico | 23.2 | 56.8 | 20 |
| Brazil | 23.8 | 56.4 | 19.8 |
| Bolivia | 28.2 | 52.2 | 19.5 |
| Costa Rica | 18.8 | 61.8 | 19.4 |
| Ecuador | 13.1 | 68.4 | 18.5 |
| Dominican republic | 34.3 | 47.6 | 18.1 |
| Panama | 15.9 | 66.4 | 17.8 |
| Venezuela | 19.5 | 63.9 | 16.6 |
| Paraguay | 10.4 | 73.5 | 16.1 |
| Uruguay | 54.4 | 30.1 | 15.5 |
| Guatemala | 20.2 | 66.3 | 13.6 |
| Honduras | 12.8 | 74.3 | 12.9 |
| Nicaragua | 30.1 | 57.8 | 12.1 |
| Total | 20.6 | 59.8 | 19.5 |

Source: Latinobarometer 2020.

Examining these data in more detail, it can be seen that among the countries with the highest proportion of disaffected citizens (bearing in mind that this refers to those who are interested in politics but who have lost trust in institutions) are some of the countries that have been at the forefront of important social protest movements in recent years, as in the cases of Peru, Colombia, Ecuador, and Chile, where there are also a high number of critical citizens. Conversely, in terms of the proportion of satisfied citizens, the case of Uruguay is a notable anomaly compared to other countries, with a value of 54.4%, which far exceeds the total average of 20.6%, and which is followed at some distance by the Dominican Republic (34.3%) and Nicaragua (30.1%).

Having set out the categorisation of citizen types, it is worth identifying their different forms of political participation, in order to confirm the hypothesis in relation to the activation of different types and degrees of both conventional and non-conventional political participation.

The data clearly show that political disaffection triggers a greater level of political participation in all its forms. In this sense, the disaffected talk about politics with their

friends by far the most, they do the most to solve problems in their communities, and are most likely to try to convince others of their political opinions, and to work for a political party or candidate (Figure 3).

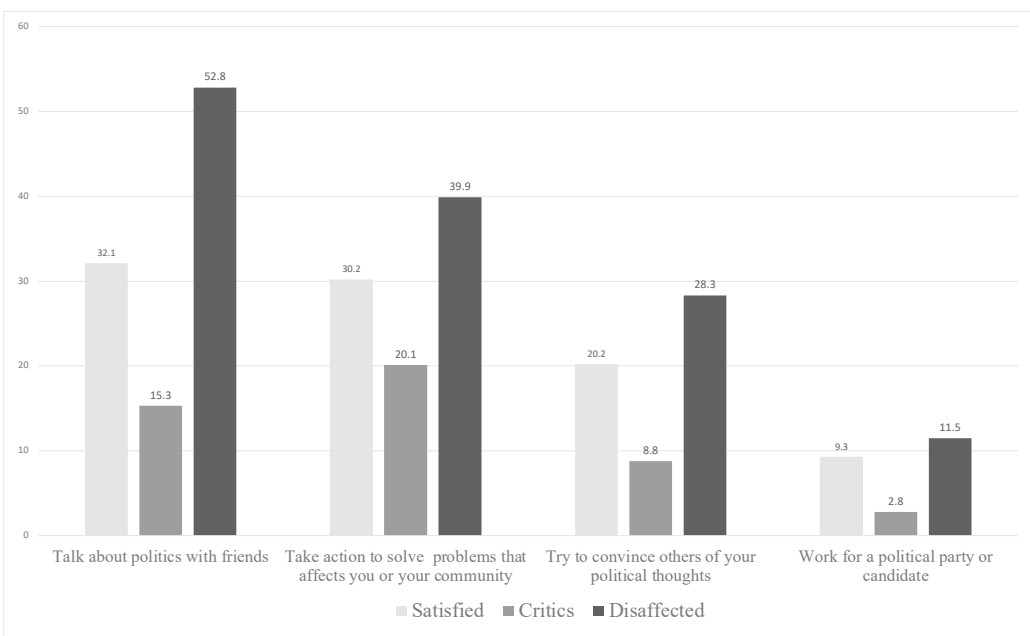

**Figure 3.** Frequency of political participation: (Very frequently/frequently). Source: Latinobarometer 2020.

Similarly, the disaffected are most likely to have signed a petition or taken part in authorised protests, and also—to a lesser extent—to have taken part in non-conventional actions, such as unauthorised protests, blocking traffic, occupying buildings, or looting (Figure 4).

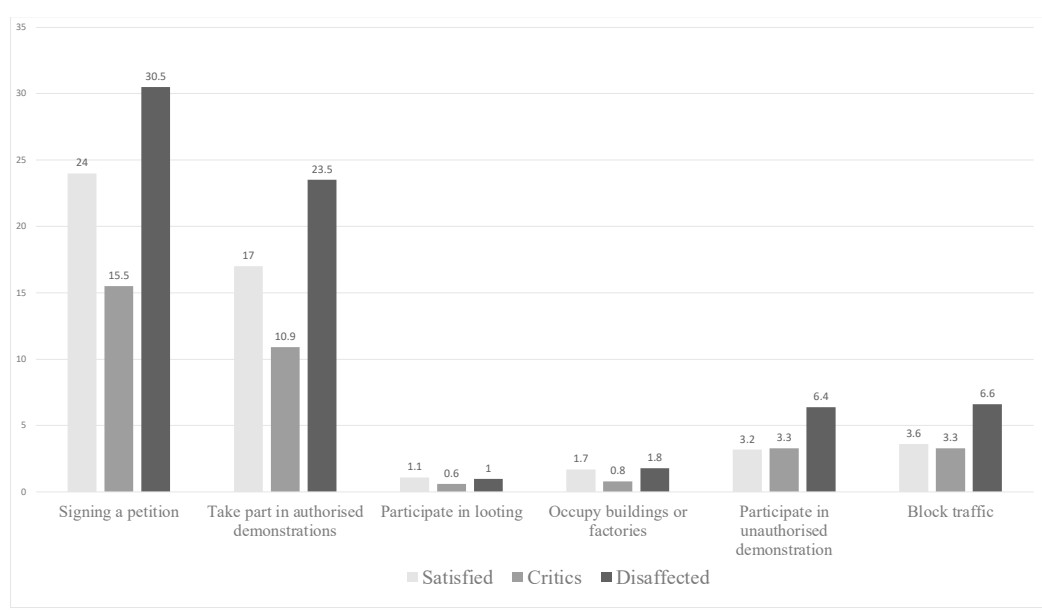

**Figure 4.** Political actions: (Have ever done). Source: Latinobarometer 2020.

Continuing this logic, the expectation should be that the disaffected would also have higher levels of digital political participation, something which must be understood in the context of the varying views held by different citizen groups in relation to the role of social networks in political participation.

In this respect, it is satisfied citizens who believe most strongly that social networks aid political participation, while critics think that they do not help political participation at all. For their part, the disaffected—despite being the group who use them the most—are also highly sceptical, inasmuch as they believe that social networks give only an illusion of real participation (Figure 5). In addition, by bringing the Latin American region into the optimum scale analysis, it can be seen that Southern Cone and Andean countries align most closely with a profile of disaffection, with the latter aligning most with critical and disaffected citizenship types. Conversely, citizens of Central American countries are the most satisfied, and have the fewest sceptics in relation to the possibility of participating politically via social networks.

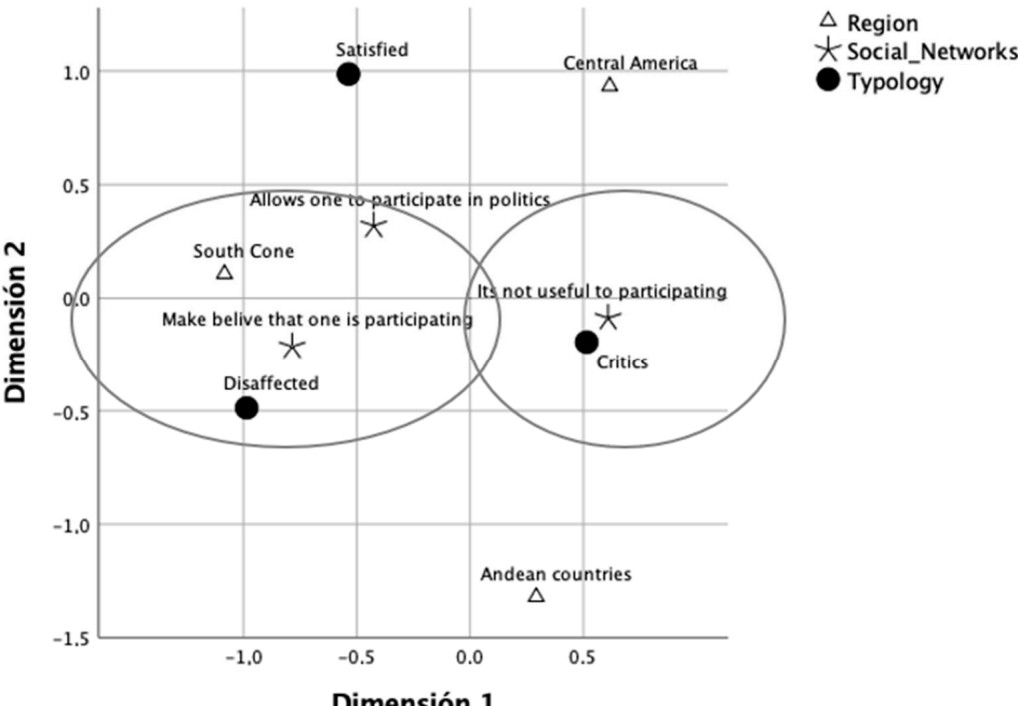

**Figure 5.** Profiles of disaffection by region and perception of political information on social networks. Source: Latinobarometer 2020.

Finally, in relation to the specific use of social networks by citizenship type, correspondence analysis confirms the notion that the advent of the disaffected citizen is associated with greater use of social networks in general, and particularly of Twitter and Instagram (Figure 6), which, as previously noted, are those most associated with uses relating to digital political participation. Conversely, critics tend to align more with the general social networks, such as WhatsApp or YouTube, and tend not to use more specific networks, such as Twitter or Instagram. Lastly, satisfied citizens are by far the least likely to use social networks of any kind. These data confirm the hypothesis that critical and disaffected attitudes, particularly the latter, are associated with an increase, not only in levels of conventional and non-conventional political participation, but also in digital participation and use of social networks as online vehicles of participation.

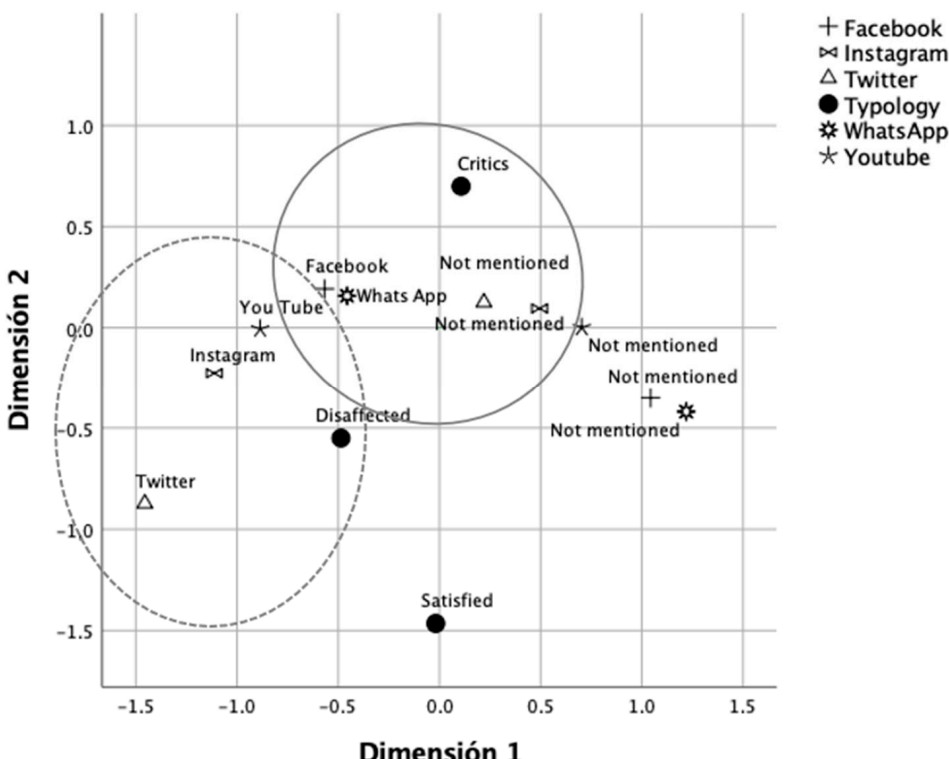

**Figure 6.** Social network following and citizen type. Source: Latinobarometer 2020.

## 6. Discussion

From a dialectical perspective, we are interested here in discussing the possible implications of the data analysed in relation to understandings of the relationship between disaffection and political participation via social networks and the internet.

The first issue to bear in mind is how connectivity rates in Latin America have evolved, inasmuch as this constitutes a precondition for analysing data relating to digital participation among citizens. In this regard, the most recent data (Latinobarometer 2020) confirm differences between countries and social strata. At a regional level, only 49.3% of citizens in Latin America have an internet connection at home. Countries with higher levels of connectivity are Argentina (86.6%), Costa Rica (74.1%), Brazil (71%), and Uruguay (67.3%). At the other end of the scale are Nicaragua (17.8%), Guatemala (24.6%), Venezuela (29.8%), and Paraguay (32%). The digital gap is especially marked in relation to economic status. For example, in 2020, just 29.7% of Latin American citizens who struggle to ensure their salary lasts until the end of the month have internet access at home, whereas, for those whose salaries allow them to save, this figure increases to 73.2%. This pattern is also repeated with similar values if the sociodemographic variable used is the poverty scale.

According to the so-called slacktivism thesis [42], the use of the internet and social networks for online political activism does not mean this is translated directly into the mobilisation of offline political actions, yet this thesis is not supported by the results of the present descriptive analysis.

On the contrary, our data point instead to the reinforcement thesis [40,41] (Schlozman et al.), an approach which suggests that digital political activism is predominantly used and monetised by citizens with a higher educational and economic status, since they have the digital ability and knowledge necessary to use the internet with political vision, as borne out of multiple correspondence analyses of citizen types, where the disaffected are those with the highest levels of political participation, and higher education and income levels (Figure 7).

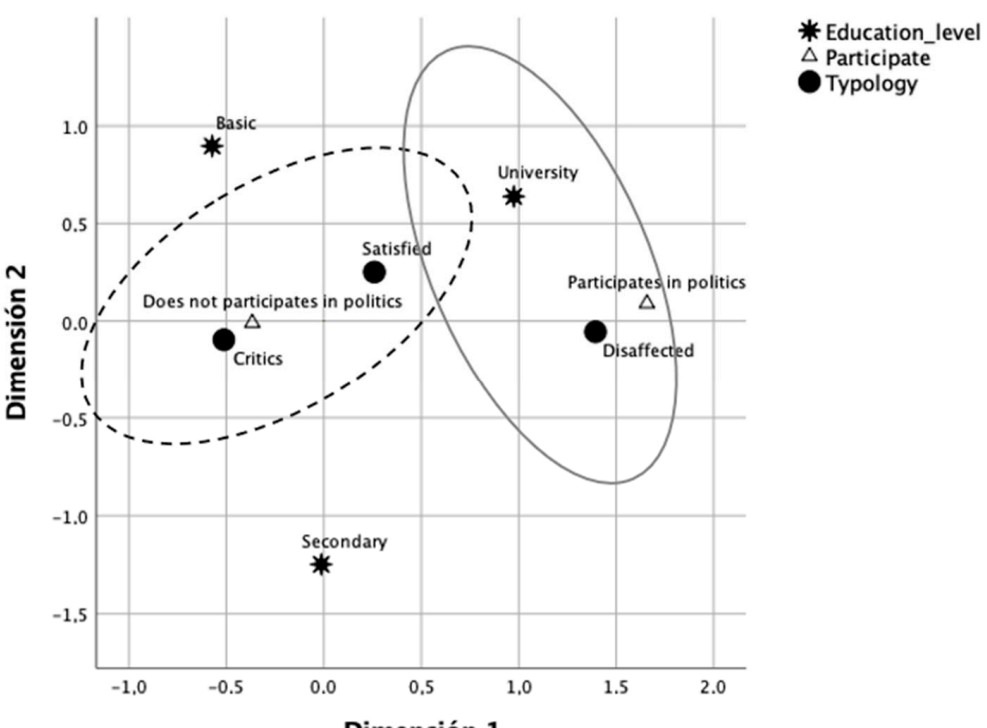

**Figure 7.** Citizen type, education level, subjective social class, and digital political participation (multiple correspondence analysis). Source: Latinobarometer 2020.

Turning to the mobilisation thesis [38], the data suggest that there is a relationship between individual political behaviour and digital political behaviour, as can be seen in Figure 8, which shows that the highest percentage of activity across all social networks is among those who participate in politics, and the disaffected.

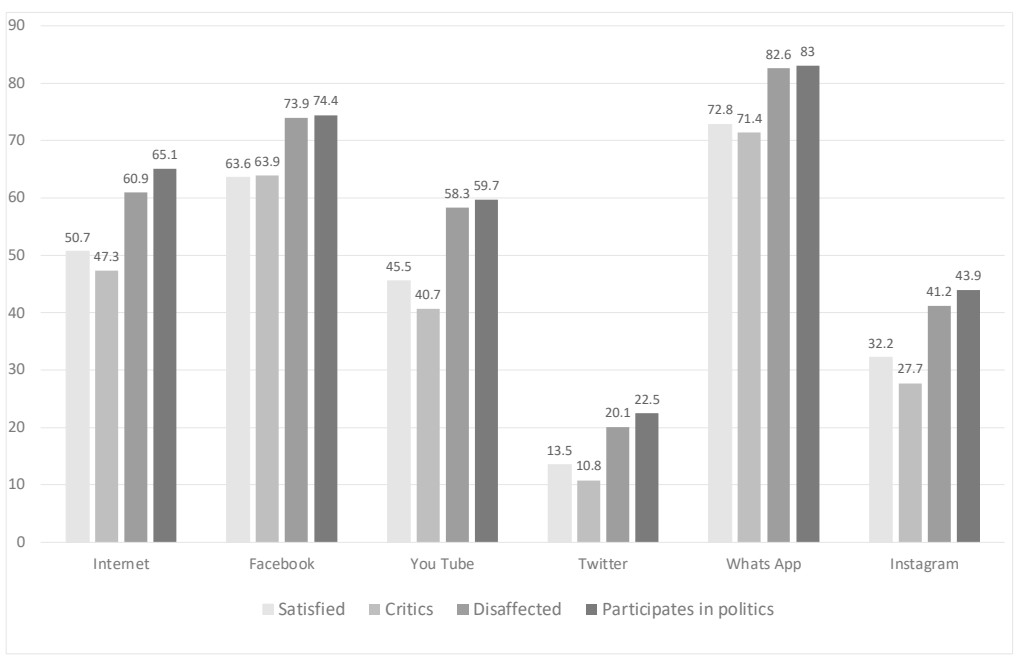

**Figure 8.** Following of social networks and citizen type. Source: Latinobarometer 2020.

## 7. Conclusions

This paper has provided a systematic analysis of political behaviour and participation and its counterpart, disaffection, from a traditional perspective, rooted theoretically in the

American functionalism of the 1950s. Changes which have taken place in the fields of communications and information over the last 25 years have led to transcendental transformations in traditional understandings of communication. Positions on the classic triangular base of political communication of "sender" (political/institutional actors)—"medium" (mass media)—"receiver" (audiences/public/electorates) have been swapped as a result of technological changes in communication and information.

As Manuel Castells [12] indeed observed some years ago, messages on digital platforms are often created for the purpose of being amplified by the media; in the process, this digitalisation of communication throws the classic communication triangle off balance. According to Castells, this signals a shift away from mass communication towards mass self-communication. This type of communication emerged through the appearance and development of Web 2.0 and Web 3.0, or, in other words, "from the group of technologies, devices and applications which support the proliferation of social spaces on the internet". In this context, political information is no longer the exclusive preserve of the mass media; rather, we are immersed in a new ecosystem which has fostered and driven a change of role, whereby citizens who traditionally merely received information, also become its creators and transmitters [14]. The mass media no longer have total control over information; rather, they now share this with social networks. Thus, already by 2018, an average of 36% of those consuming news in 37 countries did so mainly through Facebook, with this figure rising to 48% in Spain [19].

As a result, political actors have long been aware that new technologies have undoubtedly altered the political landscape, especially since the internet strategy used in Barack Obama's 2008 presidential campaign in the US. Nevertheless, it should be stressed, as already highlighted by many communications analysts, and as experience shows, that, while social networks have meant innovation in political information, they are not radically or definitively replacing journalism or conventional media. In sum, the findings of this paper corroborate the argument that social networks have led to innovation in political information and participation, but they are not replacing journalism or conventional media.

In this context, based on data provided by the Latinobarometer in its 2008, 2013, 2016, and 2020 editions, and on the citizenship typology proposed by Montero, Navarrete, and Sanz [58], the paper has analysed the phenomenon of disaffection in Latin America. This categorisation distinguishes between civies (uninterested and trusting citizens), deferentials (interested and trusting), critics (uninterested and untrusting), and the disaffected (interested and untrusting). For the purposes of the current analysis, these four categories were reduced to three: satisfied, critics, and disaffected. This has allowed a detailed analysis of the interconnected relationship between citizenship type, digital political participation, and consumption of digital media, especially social networks, in Latin America.

It is clear that there has been an explosive evolution and development of social network use in these countries, especially over the last 10 years. Social networks were divided into two groups, the first comprising networks focused on general information-sharing and instant communication (WhatsApp, Facebook, YouTube), and the second including networks that are much more specific and aligned to uses linked more to active political participation and information (Instagram and Twitter).

Taking "trust in congress/national parliament" as a category of analysis on one hand, and levels of satisfaction with democracy on the other, results from this study clearly show a continuing erosion of political and institutional trust. In terms of the economic situation, which should be analysed in the context of associated impacts of the COVID-19 pandemic over the last two years, and using the same classification, 59.8% of citizens are critics, compared to 20.6% who are satisfied and 19.8% who are disaffected. Specifically, within the last two categories—critics and disaffected—particularly notable are the cases of Peru, Colombia, Ecuador, and Chile. Conversely, the case of Uruguay is notable for its high level of satisfied citizens (54.4%), a figure far beyond the average for Latin America at 20.6%. Together with Uruguay, the Dominican Republic also stands out in relation to levels of satisfaction, with 34.3% of citizens falling into this category.

Conversely, data from our analysis show that, in general, disaffected citizens are the most active, and participate most in social networks. In a geographical distribution of this group, it can be seen that Southern Cone and Andean countries are most closely aligned with the disaffected and critical categories, while citizens of Central American countries are more satisfied and less sceptical in terms of the possibilities of social network participation.

To support the reinforcement thesis—following the categorisation by Norris [38]—would require that digital activism in Latin America be used mainly by citizens of high socio-economic and cultural status. This is indeed confirmed here through the multiple correspondence analyses of citizenship profile, whereby it is the disaffected who participate most in politics, and who also have a higher level of education and income.

In sum, and within the political and economic context resulting from the Great Recession and COVID-19, trust in institutions has been strongly eroded in recent years in the countries under study, increasing the spread of a critical and disaffected citizenry. It is precisely these categories that are associated with a type of citizenship that is much more participatory or less disaffected. Yet, it is precisely these categories that are associated with a type of citizenship that is much more active in both conventional and non-conventional forms of political participation, including digital participation and frequent use of social networks.

Finally, the conclusions of this paper can help us understand the way, and sense, in which political disaffection facilitates or hinders political participation, something especially important in the design and implementation of political and communication strategies. In this sense, having these profiles and the components that define them, both attitudinal and sociodemographic, can help us launch campaigns, strategies, and mechanisms for political action that are much more oriented towards specific profiles of citizens. Something especially important is the case of digital political participation, a constantly changing process that requires data and analysis to be adjusted to the new types of citizenship.

**Author Contributions:** Conceptualization, Á.C.-M., J.M.-P. and M.J.H.-T.; methodology, Á.C.-M., J.M.-P. and M.J.H.-T.; formal analysis, Á.C.-M., J.M.-P. and M.J.H.-T.; investigation, Á.C.-M., J.M.-P. and M.J.H.-T.; data curation, Á.C.-M.; writing—original draft preparation, Á.C.-M., J.M.-P. and M.J.H.-T.; writing—review and editing, Á.C.-M., J.M.-P. and M.J.H.-T.; visualization, Á.C.-M., J.M.-P. and M.J.H.-T.; funding acquisition, Á.C.-M., J.M.-P. and M.J.H.-T. All authors have read and agreed to the published version of the manuscript.

**Funding:** This research received no external funding.

**Institutional Review Board Statement:** Not applicable.

**Informed Consent Statement:** Not applicable.

**Data Availability Statement:** Not applicable.

**Conflicts of Interest:** The authors declare no conflict of interest. The funders had no role in the design of the study; in the collection, analyses, or interpretation of data; in the writing of the manuscript; or in the decision to publish the results.

## Notes

1    In December 2022, Twitter's new CEO, Elon Musk, confirmed his intention to increase Twitter's character limit from 280 to 4000 characters.

2    CAF Digital Ecosystem Development Index values.

3    Latin term used by Montero et al. [58]. It can be loosely translated as "affected citizens".

4    Multiple correspondence analysis (MCA) is an extension of correspondence analysis (CA) which allows one to analyse the pattern of relationships of several category-dependent variables. As such, it can also be seen as a generalisation of principal component analysis when the variables to be analysed are categorical instead of quantitative. MCA is used to analyse a set of observations described by a set of nominal variables. A series of transformations allow for the computing of the coordinates of the categories of the qualitative variables, as well as the coordinates of the observations in a representation space that is optimal for a criterion based on inertia. In the case of MCA, one can show that the total inertia is equal to the average number of categories minus one.

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
