# Peer review of "Political Disaffection and Digital Political Participation in Latin America: A Comparative Analysis of the Period 2008–2020"

_societies, doi:10.3390/soc13030059_

Round 1

Reviewer 1 Report

This paper looks at political activity in light of political disaffection in Latin American countries. It uses a reliable database and makes good use of its findings. It sets up the literature nicely, and explicates theory well enough. It separates the population into four (and then three) categories on two dimensions.

The manuscript largely succeeds. I would recommend publication with the following suggested revisions—all of which are very light, easily accomplished, and would be a good input-versus-output contribution to the final paper.

First, I think the paper needs to be more explicit about why these findings are worthwhile. Wouldn’t we expect folks who are interested in politics to talk about politics more with their friends than folks who are not interested in politics? Same for taking action to solve problems or working for a political candidate? Perhaps more pointedly, the authors do not do a very good job looping social media back into the discussion. And that leads to my second point:

Second, the figures are poor. I prefer regression tables, complete with explanation of the DV at the bottom, coefficients, and little stars telling me which variables are significant. As it stands, the graphs are confusing. It probably took some time to come up with them—and one should be proud that they can master that type of software. But when displayed in the paper, they just don’t work.

Third, the literature review section needs to be separated from the theory section. The manuscript does a good job laying out the political lay of the land, as well as the scholarly literature on activity and disaffection. There is then, however, somewhat of a harsh transition to what the authors intends to contribute. I recommend separating them, and even putting each under separate sub-headings (perhaps “Literature Review” and “Theory”…we need not overcomplicate!). In this way, the authors can more clearly set aside their contribution, and readers can more easily turn to the part of the paper where they can find what this paper is trying to uncover.

Fourth, in presenting the four citizen types, there needs to be more of an explanation for why cives and deferentials are grouped together. Other than citing another scholar (and not even giving his/her explanation), we are left with nothing. I actually think there is quite a bit of difference between folks who are interested and disinterested! Indeed, the authors seem to make that distinction with critics and disaffected. Why is the “interest” dimension thrown out for the other half of the equation? Is that really fair to the research question? There needs to be 1-2 paragraphs about this.

Fifth, and this is very easy: present a two-by-two table of cives, deferentials, critics, and disaffected. Have the dimensions be trust and interest and place them in one of the four appropriate boxes. It will be much more eye-catching, appealing, and accessible for the reader if they can SEE the argument.

Allow me to be candid: if this paper is sent back my way after revisions, I would recommend that the authors compose a letter detailing what they did to meet the five points above. The authors should feel free to copy and paste from the manuscript in that letter. For example, they could say something to the effect of: “The reviewer wanted to see a 2x2 table. It has been included and looks like the following….” Or… “the reviewer wanted a paragraph on why cives and deferentials should be grouped together. That paragraph has been added and it reads: …”

I want to accept this paper. The revisions asked for are very easy.

Author Response

FIRST:

I think the paper needs to be more explicit about why these findings are worthwhile. Wouldn’t we expect folks who are interested in politics to talk about politics more with their friends than folks who are not interested in politics? Same for taking action to solve problems or working for a political candidate? Perhaps more pointedly, the authors do not do a very good job looping social media back into the discussion.

FIRST: In response to your suggestion, a final comment has been added to the conclusions in which the usefulness of these results is explained and discussed. This is the added text: “Finally, the conclusions of this paper can help us understand the way and sense in which political disaffection facilitates or hinders political participation, something especially important in the design and implementation of political and communication strategies.In this sense, having these profiles and the components that define them, both attitudinal and sociodemographic, can help us launch campaigns, strategies and mechanisms for political action that are much more oriented towards specific profiles of citizens. Something especially important in the case of digital political participation, a constantly changing process that requires data and analysis adjusted to the new types of citizenship.” 

SECOND:

The figures are poor. I prefer regression tables, complete with explanation of the DV at the bottom, coefficients, and little stars telling me which variables are significant. As it stands, the graphs are confusing. It probably took some time to come up with them—and one should be proud that they can master that type of software. But when displayed in the paper, they just don’t work.

SECOND:
The data that we present is based on multiple correspondence analysis (MCA), a methodology that is different from regression analysis. In this case, it is a multivariable classification technique that spatially locates the relationships between categories of variables. For this, we work with the inertia and mass of the variables and their categories, being very useful to graphically explain the relationships between the data. Our problem, and for this reason we would like to thank you for your comment in reviewing the manuscript, is that we have not adequately explained how this technique works. For this, we have included in the methodology section a note that justifies its use and explains its operation. Similarly, thanks to your review, we are beginning to work with more sophisticated analysis for future articles. We understand that this first approach must be more descriptive, but that we must work on these first contributions that our work presents and advance in other modeling. That is why we are finalizing a second paper, which advances on this one that we present in Societies in which we use structural equation techniques (SEM).The added note is as follows:

“Multiple correspondence analysis (MCA) is an extension of correspondence analysis (CA) which allows one to analyze the pattern of relationships of several categorical dependent variables. As such, it can also be seen as a generalization of principal component analysis when the variables to be analyzed are categorical instead of quantitative. MCA is used to analyze a set of observations described by a set of nominal variables. A series of transformations allows the computing of the coordinates of the categories of the qualitative variables, as well as the coordinates of the observations in a representation space that is optimal for a criterion based on inertia. In the case of MCA one can show that the total inertia is equal to the average number of categories minus one.”

THIRD:

The literature review section needs to be separated from the theory section. The manuscript does a good job laying out the political lay of the land, as well as the scholarly literature on activity and disaffection. There is then, however, somewhat of a harsh transition to what the authors intends to contribute. I recommend separating them, and even putting each under separate sub-headings (perhaps “Literature Review” and “Theory”…we need not overcomplicate!). In this way, the authors can more clearly set aside their contribution, and readers can more easily turn to the part of the paper where they can find what this paper is trying to uncover.

THIRD:

As I very well suggested, the two sections have been divided, now labeled “Literature Review” and “Theory”. This provides clarity and differentiation in the text.

FOURTH:

In presenting the four citizen types, there needs to be more of an explanation for why cives and deferentials are grouped together. Other than citing another scholar (and not even giving his/her explanation), we are left with nothing. I actually think there is quite a bit of difference between folks who are interested and disinterested! Indeed, the authors seem to make that distinction with critics and disaffected. Why is the “interest” dimension thrown out for the other half of the equation? Is that really fair to the research question? There needs to be 1-2 paragraphs about this.

FOURTH:

Regarding the reduction into three categories in the classification of citizen typologies, the previous investigations show how, in most cases, there are no relevant differences between the classification of cives and deferents, and this advises us their recoding into a single category, which would be the category of satisfied citizens.

In response to your suggestion, a comment has been added. This is the text:

“In the case of the present investigation, initially we worked with the four categories, but the final results showed, like previous investigations, that no significant differences were observed between cives and deferents and this advised us to use only one category, satisfied.”

FIFTH:

Present a two-by-two table of cives, deferentials, critics, and disaffected. Have the dimensions be trust and interest and place them in one of the four appropriate boxes. It will be much more eye-catching, appealing, and accessible for the reader if they can SEE the argument.

FIFTH:

In response to your suggestion, we added the next table: Table 1.- Citizens typology

Not interested

Interested

Institutional Trust (+)

Cives

Deferentials

Institutional Trust (-)

Critics

Disafected

 FINALLY:

It is true that the main objective, and most interesting, is the study of disaffection and its effect on political participation. This is an issue that we have been working on for a long time and that seemed complete to us with the study of the impact of social networks and political participation in the digital environment. This is the reason why we were forced to carry out this paper, all of which is that the special issue of Societies focuses on digital participation. The results show that there is a relationship between digital participation and the different types of citizens, as well as (although it is not the most relevant data) that this relationship also exists among the consumers of certain social networks.

Regardless of these partial descriptive results, and this was the objective of this paper, we continue working on a model focused exclusively on digital participation. In this sense, your review has helped us a lot, since it shows that the analysis of social networks can be somewhat less important.

We wanted to thank you for your valuable contributions, which will undoubtedly reinforce the final result of our work.

Reviewer 2 Report

The paper offers an adecuate and pertinent perspective of the topic discussed. The authors should check the following lines since there are some misspelling/errors: 

- line 36; line 550 (R.); line 556 (enter); line 584; line 596 (A.); line 650 (space); line 657; line 666; line 671. Also, note that references requiere semicolon ; between authors -> Author 1, A.B.; Author 2, C.D. 

Authors must identify and declare any personal circumstances or interest that may be perceived as influencing the representation or interpretation of reported research results. If there is no conflict of interest, please state "The authors declare no conflict of interest." 

Author Response

Your review has been included in the final text

We wanted to thank you for your valuable contributions, which will undoubtedly reinforce the final result of our work.
